# Simultaneous Detection of Porcine Respiratory Coronavirus, Porcine Reproductive and Respiratory Syndrome Virus, Swine Influenza Virus, and Pseudorabies Virus via Quadruplex One-Step RT-qPCR

**DOI:** 10.3390/pathogens13040341

**Published:** 2024-04-19

**Authors:** Yan Ma, Kaichuang Shi, Zhenhai Chen, Yuwen Shi, Qingan Zhou, Shenglan Mo, Haina Wei, Liping Hu, Meilan Mo

**Affiliations:** 1College of Animal Science and Technology, Guangxi University, Nanning 530004, China; yanmayy@163.com (Y.M.); shiyuwen2@126.com (Y.S.); 2Guangxi Center for Animal Disease Control and Prevention, Nanning 530001, China; zhouqingan1@163.com (Q.Z.); moshl_2015@126.com (S.M.); weihaina@sina.cn (H.W.); hu.liping@foxmail.com (L.H.); 3College of Veterinary Medicine, Yangzhou University, Yangzhou 225009, China; zhenhai@yzu.edu.cn

**Keywords:** porcine respiratory coronavirus (PRCoV), porcine reproductive and respiratory syndrome virus (PRRSV), swine influenza virus (SIV), pseudorabies virus (PRV), multiplex RT-qPCR, detection method

## Abstract

Porcine respiratory coronavirus (PRCoV), porcine reproductive and respiratory syndrome virus (PRRSV), swine influenza virus (SIV), and pseudorabies virus (PRV) are significant viruses causing respiratory diseases in pigs. Sick pigs exhibit similar clinical symptoms such as fever, cough, runny nose, and dyspnea, making it very difficult to accurately differentially diagnose these diseases on site. In this study, a quadruplex one-step reverse-transcription real-time quantitative PCR (RT-qPCR) for the detection of PRCoV, PRRSV, SIV, and PRV was established. The assay showed strong specificity, high sensitivity, and good repeatability. It could detect only PRCoV, PRRSV, SIV, and PRV, without cross-reactions with TGEV, PEDV, PRoV, ASFV, FMDV, PCV2, PDCoV, and CSFV. The limits of detection (LODs) for PRCoV, PRRSV, SIV, and PRV were 129.594, 133.205, 139.791, and 136.600 copies/reaction, respectively. The intra-assay and inter-assay coefficients of variation (CVs) ranged from 0.29% to 1.89%. The established quadruplex RT-qPCR was used to test 4909 clinical specimens, which were collected in Guangxi Province, China, from July 2022 to September 2023. PRCoV, PRRSV, SIV, and PRV showed positivity rates of 1.36%, 10.17%, 4.87%, and 0.84%, respectively. In addition, the previously reported RT-qPCR was also used to test these specimens, and the agreement between these methods was higher than 99.43%. The established quadruplex RT-qPCR can accurately detect these four porcine respiratory viruses simultaneously, providing an accurate and reliable detection technique for clinical diagnosis.

## 1. Introduction

Respiratory coronaviruses pose serious health risks to both humans and animals. In domestic pigs, respiratory diseases are highly significant and can cause huge economic losses [1]. With the rapid development of the pig industry in China, the breeding density of pigs continues to increase, and a variety of respiratory infectious diseases threaten the health of the animals. Of the different etiological pathogens, porcine respiratory coronavirus (PRCoV), porcine reproductive and respiratory syndrome virus (PRRSV), swine influenza virus (SIV), and pseudorabies virus (PRV) cause respiratory diseases that are characterized by fever, cough, respiratory distress, slow weight gain, and, in severe cases, death in pigs [2,3,4,5].

PRCoV, which belongs to the *Alphacoronavirus* genus of the *coronaviridae* family, is a single-strand, positive-sense RNA virus with a genome measuring about 28 kb [6]. PRCoV was first reported in Belgium in 1984, and since then, it has been widely prevalent in Europe and North America [2,7]. PRCoV has also been reported in China [8], but until now, no research has reported the epidemic situations of PRCoV in China. PRCoV is thought to have originated from transmissible gastroenteritis virus (TGEV) [9,10], and there is a 621–681 nucleotide deletion at the N-terminal end of the S gene compared with TGEV, which is usually used to distinguish PRCoV from TGEV [8,11]. Currently, the number of published PRCoV strains is still limited, and the hazard of PRCoV to pig herds requires further evaluation.

PRRSV, which belongs to the *Betaarterivirus* genus of the *Arteviridae* family, is a single-strand, positive-sense RNA virus with a genome measuring about 15 kb. PRRSV is divided into two genotypes: the European genotype (genotype 1, PRRSV-1) and the North American genotype (genotype 2, PRRSV-2) [3]. PRRSV was first reported in the United States in 1987 and now has a worldwide distribution [12]. PRRSV was first reported in China in 1996, followed by widespread dissemination and rapid evolution, and both PRRSV-1 and PRRSV-2 are currently prevalent in China [13,14]. Continuous monitoring of PRRSV in pig herds remains an important defense and control route for this disease in China.

SIV, which is an influenza A virus in the *Orthomyxoviridae* family, is a single-strand, negative-sense RNA virus with a genome measuring about 13.6 kb. SIV was first recognized as a porcine respiratory pathogen in 1928 and is commonly endemic in pig farms in various countries. H1N1, H1N2, and H3N2 are the main prevalent subtypes of SIV worldwide [15,16]. Swine influenza exhibits high morbidity and low mortality, but leads to high mobility and mortality when co-infections occur [4,17]. In addition, it has been reported that pigs act as mixing vesselsfor the production of novel recombinant influenza viruses capable of replication and transmission to humans [15,18]. Therefore, SIV is a significant hazard in the pig industry and has potential public health significance.

PRV, which belongs to the *Varicellovirus* genus of the *Herpesviridae* family, is an enveloped, linear, double-stranded DNA virus with a genome measuring about 145 kb. PRV causes acute high-contact infections in a wide range of domestic and wild animals and has a significant impact on the health of the pig industry [19]. PRV was first reported in Hungary in 1902, and the first case of PRV infection in China was reported in the 1950s [19]. Variant PRV strains have severely affected pig farms in China since 2011 [20,21], causing huge economic losses. In addition, the isolation of PRV strains from human encephalitis cases has been reported, with important zoonotic potential [22,23].

PRCoV, PRRSV, SIV, and PRV infections cause respiratory symptoms, resulting in difficulty differentiating them, based only on their clinical manifestations. Unfortunately, PRCoV, PRRSV, SIV, and/or PRV co-infections often occur in pig herds, leading to complications in the diagnosis and treatment of these diseases [24,25]. In particular, PRCoV and SIV co-infections enhance the clinical signs and lung lesions in infected pigs [26], and PRRSV and PRCoV co-infections exacerbate viral infections and clinical processes [27]. Therefore, it is vital to develop a method for the detection and differentiation of these pathogens in order to sensitively and accurately diagnose these diseases. Multiplex real-time fluorescence quantitative PCR (qPCR) can detect several pathogens simultaneously in a single reaction system, with the advantages of rapidity, sensitivity, and accuracy, along with a low risk of contamination [28,29]. Compared with ordinary PCR, multiplex qPCR saves both time and costs related to the repeated detection of different pathogens and provides simplicity of operation [28,29]. To date, no method has been reported for the simultaneous detection and differentiation of PRCoV, PRRSV, SIV, and PRV. Therefore, the purpose of this study was to establish a multiplex one-step reverse-transcription qPCR (RT-qPCR) that can simultaneously detect and differentiate PRCoV, PRRSV, SIV, and PRV, providing a simple and time-efficient assay for the diagnosis and investigation of these respiratory pathogens.

## 2. Materials and Methods

### 2.1. Virus Strains

Vaccine strains of PRRSV (Ch-1R strain, attenuated vaccine, Lot No. 2020004; and HuN4-F112 strain, attenuated vaccine, Lot No. 2018006), PRV (Bartha-K61 strain, attenuated vaccine, Lot No. 20210417), TGEV (H strain, attenuated vaccine, Lot No. 2015010), porcine epidemic diarrhea virus (PEDV, CV777 strain, attenuated vaccine, Lot No. 2015010), porcine rotavirus (PRoV, G5-type NX strain, attenuated vaccine, Lot No. 2015010), and classical swine fever virus (CSFV, CVCC AV1412 strain, attenuated vaccine, Lot No. 2018063) were purchased from Harbin Harvac Biotechnology Co. Ltd. (Harbin, China). Vaccine strains of PRRSV (R98 strain, attenuated vaccine, Lot No. 2211007) and PRV (HB-98 strain, attenuated vaccine, Lot No. 2104008-2; HB2000 strain, attenuated vaccine, 2103007; and EA strain, inactivated vaccine, Lot No. 2102002) were purchased from China Animal Husbandry Industry Co. Ltd. (Chengdu, China). A vaccine strain of SIV (TJ strain, inactivated vaccine, Lot No. 20221003) was obtained from Keqian Biology Co. Ltd. (Wuhan, China). A vaccine strain of foot-and-mouth disease virus (FMDV, O/Mya98/XJ/2010 strain, inactivated vaccine, Lot No. 2211001) was purchased from Huapai Biotechnology Co. Ltd. (Chengdu, China). A vaccine strain of porcine circovirus type 2 (PCV2, ZJ/C strain, inactivated vaccine, Lot No. 20210496) was obtained from Qilu Animal Health Products Co. Ltd. (Jinan, China).

Clinical positive specimens of PRCoV, PRRSV (GXFS2022129 strain), African swine fever virus (ASFV), and porcine deltacoronavirus (PDCoV) were provided by our laboratory. All vaccine strains and clinical positive specimens were stored at −80 °C until use.

### 2.2. Clinical Specimens

A total of 4909 clinical specimens of nasal swabs, tracheas, larynxes, lungs, lymph nodes, tonsils, and spleens were collected from different pig farms, slaughterhouses, and harmless treatment plants in Guangxi Province of China from July 2022 to September 2023. All specimens were stored at −80 °C until use.

The 4909 samples, including 1247 nasal swab samples and 3662 tissue samples, were collected from 4909 pigs. The trachea, larynx, lung, lymph nodes, tonsils, and spleen from each pig were homogenized for detection of pathogens, and the homogenized tissue from each pig was considered as one sample. Of the 4909 samples, 270 nasal swab samples and 192 tissue samples came from pig farms, 947 nasal swab samples and 3154 tissue samples came from slaughterhouses, and 30 nasal swab samples and 316 tissue samples came from harmless treatment plants.

### 2.3. Primers and TaqMan Probes

Based on the genome sequences of PRCoV, PRRSV, SIV, and PRV published in the NCBI GenBank (https://www.ncbi.nlm.nih.gov/ (accessed on 15 April 2022)), primers and TaqMan probes were designed for the PRCoV S gene, PRRSV N gene, SIV M gene, and PRV gB gene, respectively (Table 1). The viral strains used for sequence comparison and the locations of the designed primers and probes are shown in Appendix A.

### 2.4. Extraction of Nucleic Acid

Tissue samples from tracheas, larynxes, lungs, lymph nodes, tonsils, and spleens were homogenized using an MM400 tissue homogenizer (Retsch, Haan, Germany), frozen and thawed three times, and centrifuged at 12,000 rpm/min at 4 °C for 5 min, and the supernatants were used for nucleic acid extraction.

One milliliter of PBS (pH 7.2) was added to the tube containing the nasal swab samples and vortexed for 30 s, and the supernatants were used for nucleic acid extraction. The vaccine solution was directly used for nucleic acid extraction.

Viral nucleic acids were extracted from 200 µL of the supernatants/solution using a TGuide S96 Nucleic Acid Extractor (Tiangen, Beijing, China) and a Viral DNA/RNA Extraction Kit (Tiangen, Beijing, China), and they were used immediately for the detection of PRCoV, PRRSV, SIV, and PRV or stored at −80 °C until use.

### 2.5. Generation of Standard Plasmid Constructs

RNA/DNA was extracted from a clinical PRCoV-positive specimen or a PRRSV, PRV, and SIV vaccine solution and was used as a template to amplify PRCoV, PRRSV, SIV, and PRV gene fragments via PCR using the primers in Table 1 with a One Step PrimeScript™ RT-PCR Kit (Perfect Real Time) (TaKaRa, Dalian, China). The PCR products were purified using a MiniBEST DNA Fragment Purification Kit Ver.4.0 (TaKaRa, Dalian, China), cloned into a pMD18-T vector (TaKaRa, Dalian, China), and transformed into DH5α competent cells (TaKaRa, Dalian, China). Positive clones were cultured at 37 °C overnight for 20–24 h, and plasmid constructs were extracted using a MiniBEST Plasmid Extraction Kit Ver.5.0 (TaKaRa, Dalian, China). The four standard plasmid constructs were named p-PRCoV, p-PRRSV, p-SIV, and p-PRV. The OD260/OD280 nm values of the standard plasmid constructs were measured, and their concentrations were determined using the following formula:$$Plasmids\;(copies/\mu L)=\frac{6.02\times 10^{23}\times plasmid\;concentration\times 10^{-9}}{plasmid\;length\;(bp)\times 660}$$

### 2.6. Optimization of Reaction Parameters

An assay was performed using a QuantStudio 6 qPCR system (ABI, Carlsbad, CA, USA) to obtain the optimal reaction conditions (annealing temperature, primer and probe concentrations, and reaction cycle). A reaction system with a volume of 25 μL was used to determine the optimal conditions. The experiment used 12.5 μL of 2 × One-Step RT-PCR Buffer (TaKaRa, Dalian, China); 0.5 μL of Ex Taq HS (TaKaRa, Dalian, China); 0.5 μL of PrimerScript RT Enzyme Mix (TaKaRa, Dalian, China); and 2.5 μL of a mixture of the four standard plasmid constructs (10^7^ copies of each, mixed at a ratio of 1:1:1:1), as a template; a mixture of four pairs of primers and four probes, with different final concentrations; and nuclease-free distilled water to reach a final volume of 25 μL. The following parameters were used: 42 °C for 5 min and 95 °C for 10 s; then 40 cycles of 95 °C for 5 s and 56 °C for 34 s. Different annealing temperatures (50, 52, 54, 56, 58, and 60 °C) and primer and probe concentrations (0.1, 0.2, 0.3, 0.4, and 0.5 pmol/µL) were used for amplification to obtain the optimal reaction conditions. The fluorescence signals were recorded at the end of each cycle. The optimal conditions were determined based on the minimum cycling threshold (Ct) and the maximum ∆Rn.

### 2.7. Generation of Standard Curves

The standard plasmid constructs, p-PRCoV, p-PRRSV, p-SIV, and p-PRV, were mixed together (1:1:1:1) and tenfold serially diluted to the final reaction concentration from 1.50 × 10^8^ to 1.50 × 10^2^ copies/µL. Then, 2.5 µL was taken as a template to generate multiplex RT-qPCR standard curves.

### 2.8. Analytical Specificity

The RNA and DNA of PRCoV, PRRSV, SIV, PRV, TGEV, PEDV, PRoV, ASFV, FMDV, PCV2, PDCoV, and CSFV were used as templates to analyze the specificity of the developed multiplex RT-qPCR. The standard plasmid constructs were used as positive controls, and negative clinical specimens and nuclease-free distilled water were used as negative controls.

### 2.9. Analytical Sensitivity

The standard plasmid constructs, p-PRCoV, p-PRRSV, p-SIV, and p-PRV, were mixed together (1:1:1:1) and tenfold serially diluted to final reaction concentrations from 1.50 × 10^8^ to 1.50 × 10^−1^ copies/μL. Then, they were used as templates to determine the limit of detection (LOD) of each plasmid construct. PROBIT regression analysis was used to evaluate the sensitivity of the multiplex RT-qPCR.

### 2.10. Repeatability Analysis

The repeatability of the established assay was evaluated by performing intra-assay and inter-assay tests. Mixtures of four standard plasmid constructs with final reaction concentrations of 1.50 × 10^7^, 1.50 × 10^5^, and 1.50 × 10^3^ copies/μL were used as templates. All reactions were repeated three times. The coefficient of variation (CV) was calculated to evaluate the repeatability of the developed method.

### 2.11. Test of the Clinical Specimens

The multiplex RT-qPCR was utilized to test 4909 clinical specimens collected from July 2022 to September 2023 in Guangxi Province, China. The published reference qPCR [30,31] was also used to test the above clinical specimens. The clinical sensitivity and specificity of the established assay were evaluated, as was the agreement between the detection results of these methods.

## 3. Results

### 3.1. Generation of the Standard Plasmid Constructs

The targeted fragments of the PRCoV S gene, PRRSV N gene, SIV M gene, and PRV gB gene were amplified using one-step PCR, purified, ligated into a pMD18-T vector, then transformed into DH5α competent cells. Positive clones were cultured at 37 °C overnight. Then, the plasmid constructs were extracted, and their concentrations were determined. The standard plasmid constructs p-PRCoV, p-PRRSV, p-SIV, and p-PRV had initial concentrations of 8.79 × 10^10^, 4.75 × 10^10^, 4.55 × 10^10^, and 6.11 × 10^10^ copies/µL, respectively. They were all diluted to 1.50 × 10^10^ copies/µL and used as standard positive controls during the development of the multiplex RT-qPCR.

### 3.2. Determining the Reaction Parameters

The standard plasmid constructs were used to determine the reaction conditions of the multiplex RT-qPCR, including the annealing temperature, primer concentration, and probe concentration, via an orthogonal test. The 25 µL reaction system and the parameters of the established multiplex RT-qPCR are shown in Table 2. The one-step procedure was as follows: 45 °C for 5 min, 95 °C for 10 s, and then 40 cycles of 95 °C for 5 s and 56 °C for 34 s. The fluorescence signals were recorded at the end of each cycle. A specimen with a Ct value ≤36 was determined to be positive, and a specimen with a Ct value > 36 was determined to be negative.

### 3.3. Generation of the Standard Curves

A mixture of the four standard plasmid constructs, with final reaction concentrations of 1.50 × 10^8^–1.50 × 10^2^ copies/μL, was used as a template for amplification to generate the standard curves of the multiplex RT-qPCR. The results showed that the slopes of the equations, correlation coefficients (R^2^), and amplification efficiencies (Es) were −3.042, 1.000, and 113.326% for PRCoV; −3.058, 0.999, and 112.335% for PRRSV; −3.068, 1.000, and 113.082% for SIV; and −3.237, 0.999, and 103.886% for PRV (Figure 1), indicating a good linear relationship between the initial templates and Ct values.

### 3.4. Specificity Analysis

The specificity of the established multiplex RT-qPCR was validated using the RNA and DNA of PRCoV, PRRSV, SIV, PRV, TGEV, PEDV, PRoV, ASFV, FMDV, PCV2, PDCoV, and CSFV as templates. The standard plasmid constructs were used as positive controls, while clinical negative specimens and nuclease-free distilled water were used as negative controls. The results showed that PRCoV, PRRSV, SIV, and PRV, corresponding to the VIC, FAM, CY5, and Texas Red fluorescence channels, generated positive amplification curves, while other viruses did not produce any amplification curves, indicating the strong specificity of the established multiplex RT-qPCR (Figure 2).

### 3.5. Sensitivity Analysis

The LOD of the multiplex RT-qPCR was determined using a mixture of the four standard plasmid constructs ranging from 1.50 × 10^8^ to 1.50 × 10^−1^ copies/μL (final reaction concentration). The results showed that the LOD was 1.50 × 10^1^ copies/μL for p-PRCoV, p-SIV, p-PRRSV, and p-PRV (Figure 3).

In addition, the LODs were also determined using PROBIT regression analysis employing four serial dilutions of 500, 250, 125, and 62.5 copies/reaction of p-PRCoV, p-PRRSV, p-SIV, and p-PRV. The Ct values and hit rates are shown in Table 3. The results showed that the LODs of p-PRCoV, p-PRRSV, p-SIV, and p-PRV were determined to be 129.594 (95% confidence interval (CI) of 116.689–152.319), 133.205 (95% CI of 120.653–156.152), 139.791 (95% CI of 127.124–166.855), and 136.600 (95% CI of 124.105–161.057) copies/reaction, respectively (Figure 4).

### 3.6. Repeatability Analysis

Mixtures of the four standard plasmid constructs with concentrations of 1.50 × 10^7^, 1.50 × 10^5^, and 1.50 × 10^3^ copies/μL (final reaction concentrations) were used to evaluate the repeatability. The results showed that the intra-assay CVs were 0.29–1.49% and the inter-assay CVs were 0.38–1.89% (Table 4), indicating the excellent repeatability of the assay.

### 3.7. Test Results of the Clinical Specimens

The 4909 clinical specimens were tested using the established multiplex RT-qPCR. The results showed that the positivity rates of PRCoV, PRRSV, SIV, and PRV were 1.36% (67/4909), 10.17% (499/4909), 4.87% (239/4909), and 0.84% (41/4909), respectively. The positivity rates of PRCoV + PRRSV + SIV, PRCoV + PRRSV, PRRSV + SIV, and PRRSV + PRV co-infections were 0.04% (2/4909), 0.06% (3/4909), 0.33% (16/4909), and 0.08% (4/4909), respectively (Table 5). The positivity rates of the specimens from pig farms, slaughterhouses, and harmless treatment plants were 3.46%, 0.95%, and 3.47% for PRCoV; 17.32%, 7.63%, and 30.64% for PRRSV; 5.41%, 4.46%, and 8.96% for SIV; and 2.60%, 0.24%, and 5.49% for PRV (Table 6). In addition, the positive samples were selected randomly for amplification and sequencing to confirm that they were PRCoV, PRRSV, SIV, and PRV, respectively.

Meanwhile, the 4909 clinical specimens were tested using the reported reference qPCR [30,31]. The results showed that the positivity rates of PRCoV, PRRSV, SIV, and PRV were 1.30% (64/4909), 9.80% (481/4909), 4.75% (233/4909), and 0.81% (40/4909), respectively. Compared to the detection results of the reported qPCR, the established quadruplex RT-qPCR showed clinical sensitivity and specificity values of 100% and 99.94% for PRCoV, 98.96% and 99.48% for PRRSV, 98.71% and 99.81% for SIV, and 97.50% and 99.96% for PRV, respectively (Table 7). The agreement between the developed quadruplex RT-qPCR and the reported reference methods was higher than 99.43% (Table 8). 

## 4. Discussion

China is the world leader in regards to pork production and consumption, with 699.95 million pigs slaughtered nationwide in 2022 (http://www.moa.gov.cn/ztzl/szcpxx/jdsj/2022/202212/ (accessed on 13 July 2023)). With the increasing density of pig herds in China, co-infections with multiple pathogens are common at pig farms. PRCoV, PRRSV, SIV, and PRV have become significant pathogens, leading to respiratory diseases and growth retardation in pigs. The similar clinical symptoms of these diseases lead to difficulties in field diagnosis; thus, accurate diagnosis depends on laboratory detection. In the quadruplex RT-qPCR established in this study, the specific primers and probes were designed to detect the conserved regions of the PRCoV S gene, PRRSV N gene, SIV M gene, and PRV gB gene. This method can specifically detect PRCoV, PRRSV, SIV, and PRV, without cross-reactions with the other swine viruses currently circulating in Chinese pig herds, including TGEV, PEDV, PRoV, ASFV, FMDV, PCV2, PDCoV, and CSFV. The LODs of PRCoV, PRRSV, SIV, and PRV were 129.594, 133.205, 139.791, and 136.600 copies/reaction, respectively. The intra-assay and inter-assay CVs ranged from 0.29% to 1.89%. The results showed that the developed method exhibited excellent specificity, high sensitivity, and good repeatability. To validate the clinical application of this method, a total of 4909 clinical specimens were tested using the developed assay and the reported reference qPCR [30,31], and the established assay showed clinical sensitivity and specificity values of 100% and 99.94% for PRCoV, 98.96% and 99.48% for PRRSV, 98.71% and 99.81% for SIV, and 97.50% and 99.96% for PRV, with agreement of more than 99.43% between these methods, demonstrating the clinical utility of the established quadruplex RT-qPCR.

The 4909 clinical specimens collected in Guangxi Province from July 2022 to September 2023 were tested using the developed quadruplex RT-qPCR. The positivity rates of PRCoV, PRRSV, SIV, and PRV were 1.36%, 10.17%, 4.87%, and 0.84%, respectively, indicating that these viruses were still prevalent in pig herds in Guangxi Province. According to reports from other countries, Korea found a seropositivity rate of 53.1% against PRCoV when testing 446 sera collected from 1998 to 1999 [32], 90.6% of sows slaughtered in Belgium in 1993 were seropositive for PRCoV [33], and growth-retarded and healthy piglets evaluated from 2000 to 2004 at a Japanese pig farm showed seropositivity rates of 62.5% and 95% and nucleic acid positivity rates of 50% and 20% for PRCoV [34]. Currently, no report has investigated the epidemic situation of PRCoV in Chinese pig herds, and this situation needs to be further investigated and analyzed. As for PRRSV, it has caused great economic losses in the swine industry around the world. The pig industry in the United States loses approximately USD 664 million annually due to PRRSV [35]. PRRSV has caused serious damage to the Chinese pig industry since 1996, especially since 2006, when the highly pathogenic PRRSV (HP-PRRSV) broke out [36]. From 2017 to 2021, the positivity rate of PRRSV in Guangxi Province was 4.92%-24.63%, while the overall positivity rate of PRRSV in South China was 18.82% (1279/6795), showing an increasing yearly trend from 4.92% to 25% [37]. As for SIV, according to serum surveillance in 17 provinces in China from 2016 to 2021, Eurasian avian-like H1N1, 2009 pandemic H1N1, and H3N2 subtype antibodies showed positivity rates of 24.75% (9986/40,343), 7.94% (3205/40,343), and 0.06% (24/40,343), respectively, indicating that Eurasian avian-like H1N1 is currently the dominant subtype in pig herds [38]. In addition, human SIV infections have been reported since 2018 [39,40]. These events suggest the zoonotic potential of SIV and the importance of continuous SIV surveillance. As for PRV, since 2011, PRV variant strains have appeared at pig farms in China, causing significant economic losses in the pig industry [41]. From 2012 to 2017, the PRV strains circulating in China showed a significantly different evolutionary relationship compared to those in other countries [42]. From 2017 to 2021, varying degrees of PRV infections were found at pig farms in several provinces in China, with the presence of classical strains, variant strains, and recombinant strains [43,44,45]. These results demonstrated that existing commercial vaccines could not prevent the emergence of new strains of PRV. In addition, human cases of PRV infection have been reported in China since 2017 [46,47,48], which suggests that more attention should be paid to the public threat due to PRV mutation and its cross-species transmission.

For the clinical specimens tested in this study, the positivity rates of PRCoV + PRRSV + SIV, PRCoV + PRRSV, PRRSV + SIV, and PRRSV + PRV co-infections were 0.04%, 0.06%, 0.33%, and 0.08%, respectively. This indicates that multiple-pathogen co-infections at pig farms are still an important problem in pig herds. Co-infections cause more heterogeneous responses than do single infections [24,25,49]. In particular, PRRSV causes immunosuppression [50], and co-infection of PRRSV with other pathogens can exacerbate the diseases [27]. In Germany, in 2009, a serum survey of a wild boar herd revealed the presence of PRCoV, SIV, PRRSV, and TGEV pathogens [51]. The presence of SIV and PRRSV co-infections was reported at pig farms in Colombia in 2014 [52], and co-infections with PRCoV and SIV were common in the fattening herds at one pig farm in Belgium from 1991 to 1992 [53]. Co-infections of PRRSV/PRV and PRRSV/SIV were also reported in Chinese pig herds [25,54]. The significance of co-infections of PRCoV, PRRSV, SIV, and/or PRV requires further evaluation, and the developed quadruplex RT-qPCR can provide a useful method for rapid and accurate detection and investigation of PRCoV, PRRSV, SIV, and PRV.

The quadruplex RT-qPCR established in this study enables the simultaneous detection and differentiation of PRCoV, PRRSV, SIV, and PRV, providing a useful method for the detection, diagnosis, and epidemiological investigation of these four pathogens. To the best of our knowledge, this is the first report to design specific primers/probes targeting the deletion region of the PRCoV S gene, which can specifically detect PRCoV. Even with the high homology between PRCoV and TGEV, the S gene exhibits a deletion region in PRCoV compared to TGEV [2,8,9,11], which can thus be used as the targeted region for the differentiation of these two pathogens. Therefore, the PRCoV S gene sequences obtained in Guangxi Province were compared with sequences downloaded from the NCBI GenBank (https://www.ncbi.nlm.nih.gov/ (accessed on 15 April 2022)) from other Chinese provinces and other countries around the world, and the deletion region of the S gene was selected when designing the specific primers and probe. After validation, the specific primers and probe could specifically detect PRCoV in the clinical specimens.

Since the worldwide COVID-19 pandemic, which began in 2019, the coronaviruses that cause respiratory diseases in humans and animals have attracted widespread attention all over the world [55,56]. Accordingly, PRCoV, as an important respiratory pathogen in pig herds, should receive more attention. However, there are currently no exact data on the epidemic status of PRCoV in Chinese pig herds. Consequently, the first priority is to strengthen epidemiological investigation, grasp the epidemic situation of the disease, and provide basic data and information to enable timely and effective measures for disease prevention and control. Therefore, the assay developed in this study provides useful technical support for the surveillance and investigation of PRCoV.

## 5. Conclusions

A quadruplex RT-qPCR for the simultaneous detection and differentiation of PRCoV, PRRSV, SIV, and PRV was successfully developed, showing excellent specificity, high sensitivity, and good repeatability. This method can simultaneously detect PRCoV, PRRSV, SIV, and PRV in a single reaction, providing a reliable and effective method for the clinical detection and differentiation of these four pathogens. The deletion region of the PRCoV S gene can be used as the targeted region for the specific detection of PRCoV using RT-qPCR. In addition, PRCoV, PRRSV, SIV, and PRV remain prevalent in southern China, and co-infections of these pathogens are an important problem that cannot be neglected.

## Figures and Tables

**Figure 1 pathogens-13-00341-f001:**
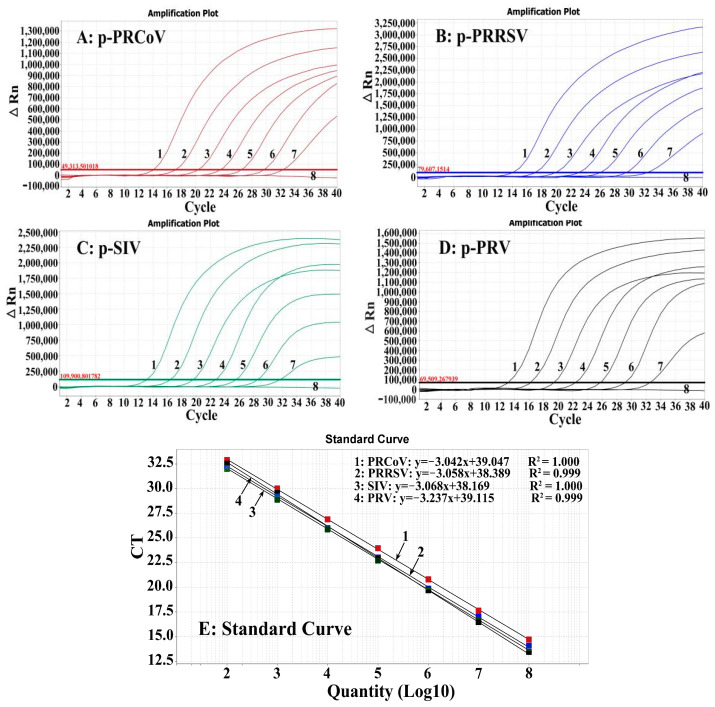
The standard curves of the multiplex RT-qPCR. (**A**–**D**): Amplification curves of the standard plasmid constructs p-PRCoV (**A**), p-PRRSV (**B**), p-SIV (**C**), and p-PRV (**D**) at final reaction concentrations ranging from 1.50 × 10^8^ to 1.50 × 10^2^ copies/μL. (**E**) Standard curves.

**Figure 2 pathogens-13-00341-f002:**
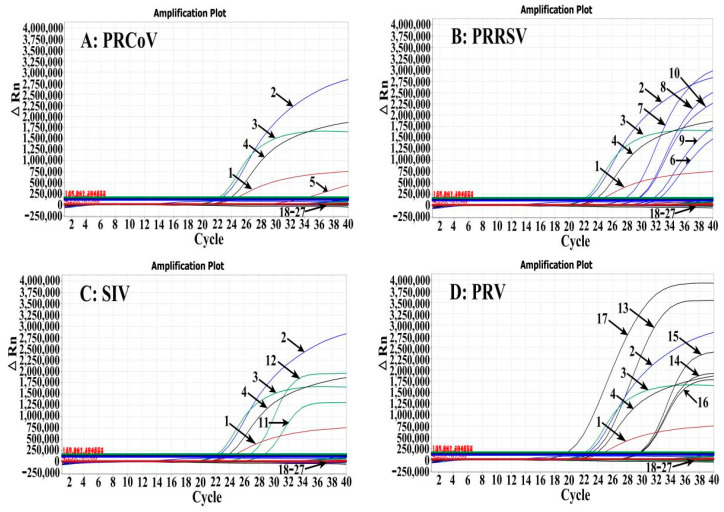
Specificity analysis. The quadruplex RT-qPCR can specifically detect PRCoV (**A**), PRRSV (**B**), SIV (**C**), and PRV (**D**). 1: p-PRCoV; 2: p-PRRSV; 3: p-SIV; 4: p-PRV; 5: PRCoV; 6: PRRSV; 7: PRRSV CH-1R strain; 8: PRRSV HuN4-F112 strain; 9: PRRSV R98 strain; 10: PRRSV GXFS2022129 strain; 11: SIV; 12: SIV TJ strain; 13: PRV; 14: PRV Bartha-k61 strain; 15: PRV HB-98 strain; 16: PRV HB2000 strain; 17: PRV EA strain; 18: TGEV H strain; 19: PEDV CV777 strain; 20: PRoV G5-type NX strain; 21: FMDV O/Mya98/XJ/2010 strain; 22: PCV2 ZJ/C strain; 23: CSFV CVCC AV1412 strain; 24: ASFV; 25: PDCoV; 26: clinical negative sample; 27: nuclease-free distilled water.

**Figure 3 pathogens-13-00341-f003:**
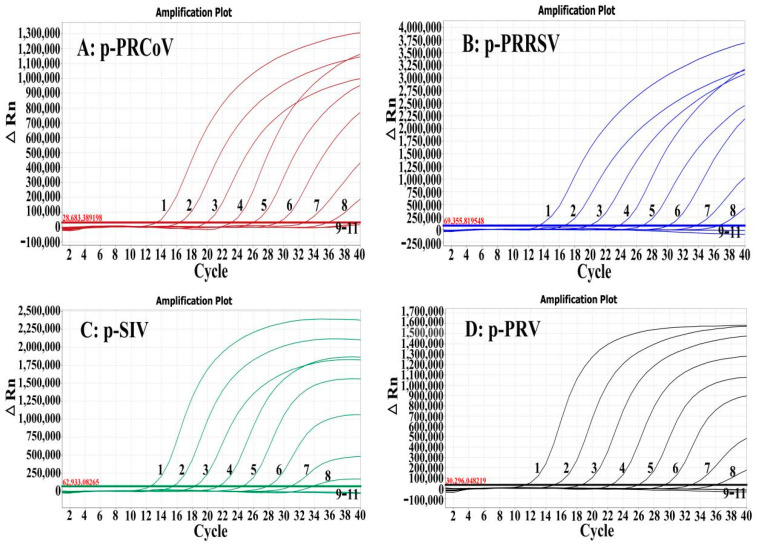
Sensitivity analysis. The amplification curves generated from the standard plasmid constructs p-PRCoV (**A**), p-PRRSV (**B**), p-SIV (**C**), and p-PRV (**D**). 1–10: The final reaction concentrations ranged from 1.50 × 10^8^ to 1.50 × 10^−1^ copies/μL. 11: Nuclease-free distilled water.

**Figure 4 pathogens-13-00341-f004:**
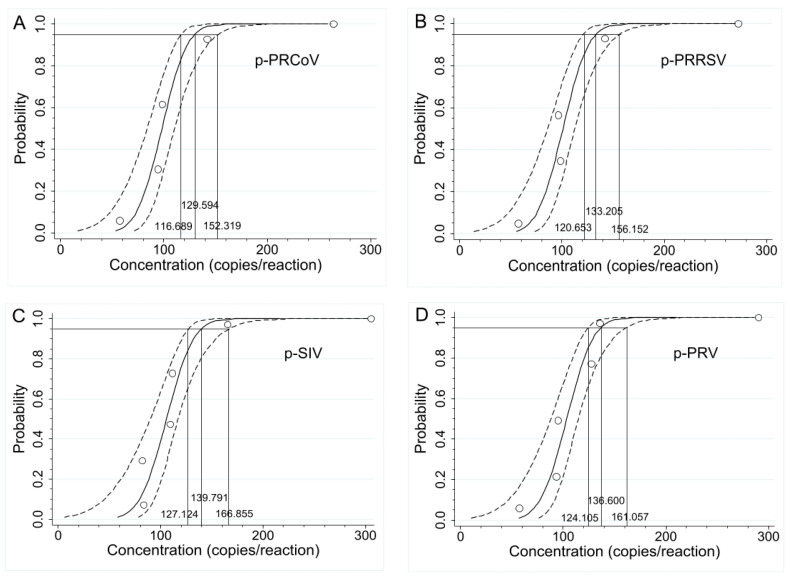
Determination of the sensitivity using PROBIT regression analysis. The LODs of p-PRCoV (**A**), p-PRRSV (**B**), p-SIV (**C**), and p-PRV (**D**) were determined to be 129.594 (95% CI of 116.689–152.319), 133.205 (95% CI of 120.653–156.152), 139.791 (95% CI of 127.124–166.855), and 136.600 (95% CI of 124.105–161.057) copies/reaction, respectively.

**Table 1 pathogens-13-00341-t001:** The designed primers and TaqMan probes.

Virus	Gene	Primer/Probe	Sequence (5′ → 3′)	Length (bp)
PRCoV	S	PRCoV-F	TGGTTGTAATGCCATTG	85
		PRCoV-R	GCCACATAACTAGCACA	
		PRCoV-P	VIC-CCTACTTCTGTAGTTTCYAATTGCACTG-BHQ1	
PRRSV	N	PRRSV-F	CCTCGTGYTGGGYGGCA	213
		PRRSV-R	GCTTCTCMGGSTTTTTCTT	
		PRRSV-P	FAM-TGGCCAGCCAGTCAATCARCTGTG-BHQ1	
SIV	M	SIV-F	CAAGACCAATCYTGTCACCTCT	91
		SIV-R	CGTCTACGCTGCAGTCC	
		SIV-P	CY5-TTCACGCTCACCGTGCCCAGT-BHQ3	
PRV	gB	PRV-F	GAGGCCCTGGAAGAAGTT	131
		PRV-R	TCCTGGACTACAGCGAGAT	
		PRV-P	Texas red-ATGCCGCGCAGCAGCACCAC-BHQ2	

Note: M = C + A, R = A + G, S = C + G, and Y = T + C.

**Table 2 pathogens-13-00341-t002:** The reaction system of the multiplex RT-qPCR.

Reagent	Volume (μL)	Final Concentration (nM)
2 × One-Step RT-PCR Buffer	12.5	/
Ex Taq HS (5 U/μL)	0.5	/
PrimerScript RT Enzyme Mix	0.5	/
PRCoV-F	0.3	300
PRCoV-R	0.3	300
PRCoV-P	0.1	100
PRRSV-F	0.3	300
PRRSV-R	0.3	300
PRRSV-P	0.2	200
SIV-F	0.2	200
SIV-R	0.2	200
SIV-P	0.3	300
PRV-F	0.3	300
PRV-R	0.3	300
PRV-P	0.2	200
Nucleic Acid Template	2.5	/
Nuclease-Free Water	Up to 25	/

**Table 3 pathogens-13-00341-t003:** The Ct values and hit rates of the serially diluted plasmid constructs.

Plasmid Construct	Concentration (Copies/Reaction)	Number of Samples	Quadruplex RT-qPCR	
			Ct (Average)	Hit Rate (%)
p-PRCoV	500	24	34.25	100
	250	24	34.84	100
	125	24	35.36	95.83
	62.5	24	ND	0
p-PRRSV	500	24	34.37	100
	250	24	34.86	100
	125	24	35.57	91.67
	62.5	24	ND	0
p-SIV	500	24	34.13	100
	250	24	34.82	100
	125	24	35.47	83.33
	62.5	24	ND	0
p-PRV	500	24	34.17	100
	250	24	34.68	100
	125	24	35.39	87.50
	62.5	24	ND	0

**Table 4 pathogens-13-00341-t004:** Repeatability analysis of the multiplex RT-qPCR.

Plasmid	Concentration (Copies/µL)	Intra-Assay Ct Value			Inter-Assay Ct Value		
		$\bar{x}$	SD	CV (%)	$\bar{x}$	SD	CV (%)
p-PRCoV	1.5 × 10^7^	16.40	0.14	0.86	16.31	0.19	1.18
	1.5 × 10^5^	23.04	0.08	0.36	23.07	0.19	0.81
	1.5 × 10^3^	29.36	0.22	0.74	29.30	0.23	0.80
p-PRRSV	1.5 × 10^7^	16.81	0.14	0.81	16.47	0.30	1.81
	1.5 × 10^5^	23.60	0.18	0.74	23.35	0.44	1.89
	1.5 × 10^3^	29.78	0.16	0.55	29.58	0.35	1.17
p-SIV	1.5 × 10^7^	15.21	0.07	0.44	15.53	0.19	1.20
	1.5 × 10^5^	22.28	0.16	0.72	22.26	0.08	0.38
	1.5 × 10^3^	28.80	0.28	0.96	28.73	0.25	0.86
p-PRV	1.5 × 10^7^	15.77	0.08	0.52	15.66	0.07	0.45
	1.5 × 10^5^	22.25	0.06	0.29	22.37	0.19	0.84
	1.5 × 10^3^	28.70	0.43	1.49	28.55	0.16	0.55

**Table 5 pathogens-13-00341-t005:** Test results of the clinical specimens.

Date	Number	Number of Positive Specimens							
		PRCoV	PRRSV	SIV	PRV	PRCoV + PRRSV + SIV	PRCoV + PRRSV	PRRSV + SIV	PRRSV + PRV
July 2022	74	0	15	0	0	0	0	0	0
August 2022	23	0	0	0	1	0	0	0	0
September 2022	109	0	6	0	0	0	0	0	0
October 2022	242	0	21	33	0	0	0	0	0
November 2022	283	0	89	8	0	0	0	7	0
December 2022	1563	24	29	41	0	0	0	0	0
January 2023	265	8	27	17	5	0	0	0	0
February 2023	502	0	38	28	1	0	0	0	0
March 2023	304	10	28	7	3	0	0	3	2
April 2023	174	0	20	8	2	0	0	0	0
May 2023	334	7	48	7	4	0	0	0	1
June 2023	195	6	21	6	0	0	0	0	0
July 2023	636	7	107	76	25	0	0	3	1
August 2023	189	0	34	3	0	0	0	0	0
September 2023	16	5	16	5	0	2	3	3	0
Total	4909	67	499	239	41	2	3	16	4
Positivity Rate (%)		1.36%	10.17%	4.87%	0.84%	0.04%	0.06%	0.33%	0.08%

**Table 6 pathogens-13-00341-t006:** Test results of clinical specimens from different sources.

Source	Sample	Number	Number of Positive Specimens							
			PRCoV	PRRSV	SIV	PRV	PRCoV + PRRSV	PRRSV + SIV	PRRSV + PRV	PRCoV + PRRSV + SIV
Pig Farms	Nasal swab	270	16 (5.93%)	8 (2.96%)	6 (2.22%)	1 (0.37%)	0	0	0	0
	Tissue	192	0	72 (37.5%)	19 (9.90%)	11 (5.73%)	0	3 (1.56%)	1 (0.52%)	0
	Total	462	16 (3.46%)	80 (17.32%)	25 (5.41%)	12 (2.60%)	0	3 (0.65%)	1 (0.22%)	0
Slaughterhouses	Nasal swab	947	23 (2.43%)	1 (0.11%)	32 (3.38%)	0	0	0	0	0
	Tissue	3154	16 (0.51%)	312 (9.89%)	151 (4.79%)	10 (0.32%)	0	1 (0.03%)	3 (0.10%)	0
	Total	4101	39 (0.95%)	313 (7.63%)	183 (4.46%)	10 (0.24%)	0	1 (0.02%)	3 (0.07%)	0
Harmless Treatment Plants	Nasal swab	30	7 (23.33%)	13 (43.33%)	5 (16.67%)	0	0	3 (10%)	0	2 (6.67%)
	Tissue	316	5 (1.58%)	93 (29.43%)	26 (8.23%)	19 (6.01%)	3 (0.95%)	9 (2.85%)	0	0
	Total	346	12 (3.47%)	106 (30.64%)	31 (8.96%)	19 (5.49%)	3 (0.87%)	12 (3.47%)	0	2 (0.58%)
Total		4909	67 (1.36%)	499 (10.17%)	239 (4.87%)	41 (0.84%)	3 (0.06%)	16 (0.33%)	4 (0.08%)	2 (0.04%)

**Table 7 pathogens-13-00341-t007:** The clinical sensitivity and specificity of the established quadruplex RT-qPCR.

The Established Quadruplex RT-qPCR		The Reference Multiplex qPCR		Total	Clinical Sensitivity (95% CI)	Clinical Specificity (95% CI)
		Positive	Negative			
PRCoV	Positive	64	3	67	100% (94.34–100%)	99.94% (99.82–99.98%)
	Negative	0	4842	4842		
	Total	64	4845	4909		
PRRSV	Positive	476	23	499	98.96% (97.59–99.56%)	99.48% (99.22–99.65%)
	Negative	5	4405	4410		
	Total	481	4428	4909		
SIV	Positive	230	9	239	98.71% (96.28–99.56%)	99.81% (99.63–99.90%)
	Negative	3	4667	4670		
	Total	233	4676	4909		
PRV	Positive	39	2	41	97.50% (87.12–99.56%)	99.96% (99.85–99.99%)
	Negative	1	4867	4868		
	Total	40	4869	4909		

**Table 8 pathogens-13-00341-t008:** The agreement between the established quadruplex RT-qPCR and the reported reference RT-qPCR.

Method	Positive Specimens			
	PRCoV (%)	PRRSV (%)	SIV (%)	PRV (%)
The Developed Multiplex RT-qPCR	67/4909 (1.36%)	499/4909 (10.17%)	239/4909 (4.87%)	41/4909(0.84%)
The Reference Multiplex qPCR	64/4909 (1.30%)	481/4909 (9.80%)	233/4909 (4.75%)	40/4909(0.81%)
Positive Agreement (95% CI)	100% (94.34–100%)	98.96% (97.59–99.56%)	98.71% (96.28–99.56%)	97.50% (87.12–99.56%)
Negative Agreement (95% CI)	99.94% (99.82–99.98%)	99.48% (99.22–99.65%)	99.81% (99.63–99.90%)	99.96% (99.85–99.99%)
Overall Agreement (95% CI)	99.94% (99.82–99.98%)	99.43% (99.18–99.61%)	99.76% (99.57–99.86%)	99.94% (99.82–99.98%)
Kappa	0.977	0.968	0.973	0.963

## Data Availability

The original contributions presented in the study are included in the article/Appendix A, further inquiries can be directed to the corresponding author/s.

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
