# Peer review of "Simultaneous Detection of Porcine Respiratory Coronavirus, Porcine Reproductive and Respiratory Syndrome Virus, Swine Influenza Virus, and Pseudorabies Virus via Quadruplex One-Step RT-qPCR"

_pathogens, 2024, doi:10.3390/pathogens13040341_

Round 1

Reviewer 1 Report

Comments and Suggestions for Authors

The manuscript describes the development of a multiplex PCR to detect four viral pathogens. Similar test developments are cited by the authors (references 30 and 31). 

83-84, 316-317: These statements are incorrect; these agents do not cause similar pathologic changes in the lung of affected pigs.

124-125: More detail needs to be provided regarding the characterization of the isolates used to develop the primers to ensure the reader that the test can detect various strains of these viruses (such as EU and NA PRRSV, as well as different lineages). Should be more information than just “published in GenBank.”

132: Why were liver and spleen used in the assorted tissues for testing, if this is designed as a test to be used for respiratory pathogens?

Figure 2: Consider a better way to represent this data, maybe break it up instead of having 17 different curves on the same grid.

367-368: This sentence is directly contradicted by the very low number of co-infections detected in the sentence immediately before it.  

Comments on the Quality of English Language

Many errors, mostly in grammar, spelling, tense, etc.

Author Response

Reviewer 1

Comments and Suggestions for Authors

  1. The manuscript describes the development of a multiplex PCR to detect four viral pathogens. Similar test developments are cited by the authors (references 30 and 31). 

Response:  In our study, we developed a quadruplex RT-qPCR to simultaneously detect PRCoV, PRRSV, SIV, and PRV. We used the methods in the reference 30 and 31 as the reference methods to detect PRCoV, PRRSV, PRV, and PRV. We used the singleplex RT-qPCR/qPCR in the reference 30 to detect PRCoV, PRRSV, PRV, and PRV, respectively, and used the multiplex RT-qPCR/qPCR to detect PRRSV and PRV.

  1. 83-84, 316-317: These statements are incorrect; these agents do not cause similar pathologic changes in the lung of affected pigs.

Response: The sentence has been corrected according to the review’s suggestion. Please see Lines 87-89, and Lines 336-338 in the revised manuscript.

  1. 124-125: More detail needs to be provided regarding the characterization of the isolates used to develop the primers to ensure the reader that the test can detect various strains of these viruses (such as EU and NA PRRSV, as well as different lineages). Should be more information than just “published in GenBank.”

Response: In our study, the sequences of the typical strains of PRCoV, PRRSV, SIV, and PRV were downloaded from GenBank, and compared. The conserved regions were selected as the targeted regions of primers and probes. Since there are many strains of each virus, only some typical strains were selected to compare their genome sequences. As for PRRSV, different lineages were compared, and the primers and probe were designed to detect PRRSV-1 and PRRSV-2, and different lineages of PRRSV-2. Please see Lines 134-135 and the Supplementary Figure S1.

  1. 132: Why were liver and spleen used in the assorted tissues for testing, if this is designed as a test to be used for respiratory pathogens?

Response: Liver and spleen are usually used to detect PRCoV, PRRSV, SIV, and PRV due to their high vial loads in these organs. Since the samples were collected from the dead pigs, so the snout swabs, tracheas, larynxes, lungs, lymph nodes, tonsils, and spleens were collected for detection of PRCoV, PRRSV, SIV, and PRV in this study.

In this study, only spleen was used to detect these viruses. The “liver” in the manuscript is a clerical error, and has been deleted in the revised manuscript. Please see Line 142 in the revised manuscript.

  1. Figure 2: Consider a better way to represent this data, maybe break it up instead of having 17 different curves on the same grid.

Response: According to the reviewer’s suggestion, Figure 2 was divided into four figures (A, B, C, D) in the revised manuscript. Please see Figure 2 in the revised manuscript.

  1. 367-368: This sentence is directly contradicted by the very low number of co-infections detected in the sentence immediately before it.  

Response: Even if the relatively low number of co-infections in this study, co-infections exacerbate these diseases and cause huge losses to pig herds, so the problem of co-infections cannot be neglected. Therefore, this sentence is used here. Please see Lines 391-392 in the revised manuscript.

Comments on the Quality of English Language

  1. Many errors, mostly in grammar, spelling, tense, etc.

Response: The manuscript has been undergone English language editing by MDPI. Please see the revised manuscript.

Best regards,

Kaichuang Shi

Reviewer 2 Report

Comments and Suggestions for Authors

This is an interesting study reporting a quadruplex RT-qPCR assay to simultaneously detect four important viruses in pigs. While the authors provided enough information in methodology, results, and discussion sections, the Introduction section lacks cohesiveness. I suggest authors significantly improve the Introduction section, technically and grammatically. 

1. English can be improved such as in L36: "respiratory infection is the very important diseases" and L37: "industrialized pig industry". 

2. L37-40: "With the development of large-scale and industrialized pig industry in China in recent years, the breeding density of pigs continues to increase, and a variety of respiratory infectious diseases cause a great threat to the health of pigs, leading to high morbidity rate and mortality rate in pig farms." Such sentences provide no information. Please be specific and provide citations. 

3. L38-44: In-text citations should be provided. 

4. L46: PRCoV was first reported....

5. L68: "high mobility"?

6. L313-314: "PRCoV, PRRSV, SIV, and PRV have become important factors". Viruses are not factors. 

7. L316: "pathological damages of these diseases". Please improve this sentence. 

8. L318: "judgment"?

9. L321=322: Which other viruses are reported more frequently in swine herds in China? Provide citations to support the statement. 

Comments on the Quality of English Language

English can be improved at many places. I suggest authors take help of a native English speaker. 

Author Response

Reviewer 2

Comments and Suggestions for Authors

This is an interesting study reporting a quadruplex RT-qPCR assay to simultaneously detect four important viruses in pigs. While the authors provided enough information in methodology, results, and discussion sections, the Introduction section lacks cohesiveness. I suggest authors significantly improve the Introduction section, technically and grammatically. 

  1. English can be improved such as in L36: "respiratory infection is the very important diseases" and L37: "industrialized pig industry". 

Response: These sentences have been rewritten. Please see Lines 37-41 in the revised manuscript.

  1. L37-40: "With the development of large-scale and industrialized pig industry in China in recent years, the breeding density of pigs continues to increase, and a variety of respiratory infectious diseases cause a great threat to the health of pigs, leading to high morbidity rate and mortality rate in pig farms." Such sentences provide no information. Please be specific and provide citations. 

Response: The sentence has been rewritten. Please see Lines 38-41 in the revised manuscript.

  1. L38-44: In-text citations should be provided. 

Response: The references 2, 3, 4, and 5 were cited. Please see Line 46 in the revised manuscript.

  1. L46: PRCoV was first reported....

Response: The sentence has been rewritten according to the reviewer’s suggestion. Please see Lines 48-50 in the revised manuscript.

  1. L68: "high mobility"?

Response: The word morbidity is correct. Please see Line 72 in the revised manuscript.

  1. L313-314: "PRCoV, PRRSV, SIV, and PRV have become important factors". Viruses are not factors. 

Response: The sentence has been rewritten. Please see Lines 333-336 in the revised manuscript.

  1. L316: "pathological damages of these diseases". Please improve this sentence. 

Response: The sentence has been rewritten. Please see Lines 336-338 in the revised manuscript.

  1. L318: "judgment"?

Response: The word "judgment" has been deleted. Please see Lines 338 in the revised manuscript.

  1. L321=322: Which other viruses are reported more frequently in swine herds in China? Provide citations to support the statement. 

Response: The sentence has been rewritten. Please see Lines 341-343 in the revised manuscript.

Comments on the Quality of English Language

  1. English can be improved at many places. I suggest authors take help of a native English speaker. 

Response: The manuscript has been undergone English language editing by MDPI. Please see the revised manuscript.

Best regards,

Kaichuang Shi

Reviewer 3 Report

Comments and Suggestions for Authors

The article entitled “Simultaneous Detection of Porcine Respiratory Coronavirus, Porcine Reproductive and Respiratory Syndrome Virus, Swine Influenza Virus and Pseudorabies Virus by Quadruplex One-Step RT-qPCR” written by Kaichuang Shi at all focuses on detection of mayor important viruses responsible for respiratory inflammations in pigs topic. In did, Porcine respiratory coronavirus (PRCoV), porcine reproductive and respiratory syndrome virus (PRRSV), swine influenza virus (SIV), and pseudorabies virus (PRV) are really important pathogens, which play crucial role in swine welfare, and cause major economical loses in swine industry worldwide. As authors emphasize, the simple, one-step diagnostic methods for few pathogens are limited, thus the subject of article is quite interesting.

Authors established quadruplex RT-qPCR, and proved it can accurately detect four crucial porcine respiratory viruses simultaneously, providing an accurate and reliable detection technique. Presented diagnostic technic have a high applicability potential and that fact increases the practical and novelty aspects of the research.

The construction of article is really clear. The introduction is brief and essential, material and methods are short and clear. The aim of article is supported by obtained results, presentation of results is well done, in short, essential chapters, clear and understandable. Authors described protocol step by step in details, what may simplify adapting it to clinical practice.Authors provide specificity and sensitivity analysis of designed primers, the results approved that designed method have equal diagnostic value as referential ones (line 290-293). In Discussion author compered results with literature, highlights the most important aspect of the research- accuracy and reliability of presented technique. The research is well designed. 

Author Response

Reviewer 3

Comments and Suggestions for Authors

The article entitled “Simultaneous Detection of Porcine Respiratory Coronavirus, Porcine Reproductive and Respiratory Syndrome Virus, Swine Influenza Virus and Pseudorabies Virus by Quadruplex One-Step RT-qPCR” written by Kaichuang Shi et al focuses on detection of mayor important viruses responsible for respiratory inflammations in pigs topic. In did, Porcine respiratory coronavirus (PRCoV), porcine reproductive and respiratory syndrome virus (PRRSV), swine influenza virus (SIV), and pseudorabies virus (PRV) are really important pathogens, which play crucial role in swine welfare, and cause major economical loses in swine industry worldwide. As authors emphasize, the simple, one-step diagnostic methods for few pathogens are limited, thus the subject of article is quite interesting.

Authors established quadruplex RT-qPCR, and proved it can accurately detect four crucial porcine respiratory viruses simultaneously, providing an accurate and reliable detection technique. Presented diagnostic technic have a high applicability potential and that fact increases the practical and novelty aspects of the research.

 The construction of article is really clear. The introduction is brief and essential, material and methods are short and clear. The aim of article is supported by obtained results, presentation of results is well done, in short, essential chapters, clear and understandable. Authors described protocol step by step in details, what may simplify adapting it to clinical practice. Authors provide specificity and sensitivity analysis of designed primers, the results approved that designed method have equal diagnostic value as referential ones (line 290-293). In Discussion author compered results with literature, highlights the most important aspect of the research- accuracy and reliability of presented technique. The research is well designed. 

Response: Thanks very much for the high recognition from the reviewer.

Best regards,

Kaichuang Shi

Reviewer 4 Report

Comments and Suggestions for Authors

1. please indicate in detail how many out swabs, tracheas, larynxes, lungs, lymph nodes, tonsils, and spleens are in each of the 4909 samples.
2. Please add the rate of compliance of the method with international standards for various viruses and the Kappa coefficient.
3. a native English speaking professional is needed to revise the manuscript

Comments on the Quality of English Language

 a native English speaking professional is needed to revise the manuscript

Author Response

Response Notes

April 2, 2024

Dear review,

We have revised our manuscript carefully according to the reviewer's suggestions. The details are as follows.

Reviewer 4

Comments and Suggestions for Authors

  1. please indicate in detail how many snout swabs, tracheas, larynxes, lungs, lymph nodes, tonsils, and spleens are in each of the 4909 samples.

Response: The information on the 4,909 samples has been added in the revised manuscript. Please see Lines 132-138 in the revised manuscript. The content in details is as follows.

The 4,909 samples, including 1,247 nasal swab samples and 3,662 tissue samples, were collected from 4,909 pigs. The trachea, larynx, lung, lymph nodes, tonsil, and spleen from each pig were homogenized for detection of pathogens, and the homogenized tissue from each pig was considered as one sample. Of the 4,909 samples, 270 nasal swab samples and 192 tissue samples came from pig farms, 947 nasal swab samples and 3,154 tissue samples came from slaughterhouses, and 30 nasal swab samples and 316 tissue samples came from harmless treatment plants.

The detection results of the 4,909 clinical samples using the developed quadruplex RT-qPCR have been shown in Table 6 in the revised manuscript. Please see Table 6 in the revised manuscript.

  1. Please add the rate of compliance of the method with international standards for various viruses and the Kappa coefficient.

Response: Up to date, no RT-qPCR method for PRCoV is recommended by the World Organization for Animal Health (WOAH). The RT-qPCR method recommended by WOAH for detecting PRRSV was used to detect PRRSV-1 and PRRSV-2, respectively, but not to simultaneously detect PRRSV-1 and PRRSV-2. The RT-qPCR method recommended by WOAH for detecting SIV involves one forward primer, probe, and two reverse primers. The RT-qPCR mentioned by WOAH in two published references was used to distinguish PRV wild-type strains and vaccine strains.

 In this study, four pairs of specific primers and probes were used to develop quadruplex RT-qPCR for simultaneous detection and differentiation of different types and strains of PRCoV, PRRSV, SIV and PRV. The methods recommended by WOAH are unsuitable for simultaneous detection of these pathogens. Therefore, we did not use these methods in this study, but use the reported methods as reference methods for the detection of PRCoV, PRRSV, SIV and PRV.

The rate of compliance (agreement) between the developed quadruplex RT-qPCR and the reported reference methods, and the Kappa coefficient have been added to Table 8 in the revised manuscript. Please see Table 8 in the revised manuscript.

The references are as follows.

  1. Zhang, Q.; Yang, F.; Gao, J.; Zhang, W.; Xu, X. Development of multiplex TaqMan qPCR for simultaneous detection and differentiation of eight common swine viral and bacterial pathogens. Braz. J. Microbiol. 2022, 53, 359-368.
  2. Sunaga, F.; Tsuchiaka, S.; Kishimoto, M.; Aoki, H.; Kakinoki, M.; Kure, K.; Okumura, H.; Okumura, M.; Okumura, A.; Nagai, M.; et al. Development of a one-run real-time PCR detection system for pathogens associated with porcine respiratory diseases. J. Vet. Med. Sci. 2020, 82, 217-223.
  3. A native English speaking professional is needed to revise the manuscript.

Response: The manuscript has been undergone English language editing by MDPI. Please see the revised manuscript. Please see the certificate in the attachment.

Comments on the Quality of English Language

  1.  A native English speaking professional is needed to revise the manuscript.

Response: The manuscript has been undergone English language editing by MDPI. Please see the revised manuscript. Please see the certificate in the attachment.

Best regards,

Kaichuang Shi

Round 2

Reviewer 4 Report

Comments and Suggestions for Authors

1, How does this method exclude false positives that are not due to vaccine immunity? For example, how to exclude the effect of PRV gene deletion seedlings when designing primers based on the PRV gB gene. Why not design primers using the gE gene.
2. 131 rows. 12000rpm centrifugation for 15 minutes? Excessive centrifugation will reduce the sensitivity of the assay. Please refer to
3、Whether the vaccine samples used are weakly virulent or inactivated, please indicate the specific manufacturer and production batch number.

Author Response

Response Notes

April 7, 2024

Dear reviewer,

We have revised our manuscript carefully according to the reviewer #4's suggestions. The response notes are as follows in details.

Reviewer #4

Comments and Suggestions for Authors

  1. How does this method exclude false positives that are not due to vaccine immunity? For example, how to exclude the effect of PRV gene deletion seedlings when designing primers based on the PRV gB gene. Why not design primers using the gE gene.

Response: To exclude false positives in this study, the recombinant standard plasmid constructs, and the positive tissue samples were used as positive control. The negative tissue samples, and nuclease-free distilled water were used as negative controls. In addition, the positive samples detected by the developed quadruplex RT-qPCR were selected randomly for amplification and sequencing to further confirm that they were PRCoV, PRRSV, SIV, and PRV, respectively. This content has been added to the revised manuscript. Please see Lines 9-10 in the Part 3.7. Test Results of the Clinical Specimens in the revised manuscript.

The purpose of this assay was to detect both vaccine and wild strains of PRV, so the conserved gB gene, but not gE gene, was selected as the targeted region of the primers and probes. If a positive sample needs to be further identified as a vaccine or wild strain of PRV, other singleplex or multiplex qPCR can be used to further detect and differentiate them.

  1. 131 rows. 12000rpm centrifugation for 15 minutes? Excessive centrifugation will reduce the sensitivity of the assay. Please refer to.

Response: we are very sorry that this is a clerical error. “15 minutes” has been changed to “5 minutes”. Please see Line 3 in the Part 2.4. Extraction of Nucleic Acid in the revised manuscript.

  1. Whether the vaccine samples used are weakly virulent or inactivated, please indicate the specific manufacturer and production batch number.

Response: The information on the vaccine samples has been added in more details in the revised manuscript, including attenuated or inactivated vaccine, and the specific manufacturer and production batch number. Please see Lines 1-17 in the Part 2.1. Virus Strains in the revised manuscript.

Best regards,

Kaichuang Shi

Round 3

Reviewer 4 Report

Comments and Suggestions for Authors

 good job